# Nucleophilicity at copper(-I) in a compound with a Cu–Mg bond

Ross A. Jackson[1], Nicholas J. Evans [1], Dawid J. Babula[1,2], Thomas M. Horsley Downie [1], Rex S. C. Charman[1], Samuel E. Neale [1], Mary F. Mahon[1] & David J. Liptrot [1] ✉

Copper is ubiquitous as a structural material, and as a reagent in (bio)chemical transformations. A vast number of chemical reactions rely on the near-inevitable preference of copper for positive oxidation states to make useful compounds. Here we show this electronic paradigm can be subverted in a stable compound with a copper-magnesium bond, which conforms to the formal oxidation state of Cu(-I). The Cu-Mg bond is synthesized by the reaction of an N-heterocyclic carbene (NHC) ligated copper alkoxide with a dimeric magnesium(I) compound. Its identity is confirmed by single-crystal X-ray structural analysis and NMR spectroscopy, and computational investigations provide data consistent with a high charge density at copper. The Cu-Mg bond acts as a source of the cupride anion, transferring the NHC-copper fragment to electrophilic s-, p-, and d-block atoms to make known and new copper-containing compounds.

Copper is a ubiquitous element, showing up everywhere from the Statue of Liberty to biological systems[1]. Its synthetic pedigree cannot be understated; copper catalysis is important in bulk and fine chemicals applications, ranging from the industrial scale synthesis of methanol[2], to the Nobel prize winning alkyne-azide click reaction[3,4]. This chemical convenience relies heavily on the consistent electronic nature of copper; homogeneous copper compounds reliably occur in oxidation states of +1 and +2, and less commonly in higher oxidation states[1]. In all such cases copper is electrophilic in character, and stable, nucleophilic copper complexes adopting oxidation states below zero remain unknown. A few isolable compounds which can be assigned such a lower oxidation state have been reported[5–10], one class being those where copper is bonded to a more electropositive element. For example, the groups of Hill and McMullin[11]; and Zhao, Frenking, Goicoechea and Aldridge[12] reported compounds containing covalent Cu–Al bonds. In their reactions with hetero-cumulenes, these bonds showed some evidence of nucleophilicity at copper. This behavior was, however, inconsistent between complexes and analysis of these and related coinage metal complexes indicated that the Cu–Al bond can, at best, be thought of as

ambiphilic[13–15]. This lacuna is in stark contrast to the other late transition metals, where sub-zero oxidation state complexes are increasingly common[16].

Compounds containing metal-metal bonds continue to attract significant research interest, and a substantial swathe of the potential homometallic metal-metal bonds are now reported[17]. These species have revolutionized chemistry, changing how bonding is rationalized[18,19]; our understanding of the underpinning forces in molecular structures[20,21]; and providing some of the most exciting new examples of reactivity of the 21st century[22–24]. In contrast, the much larger range of possible heterobimetallic metal-metal bonds are far less well-explored. One significant advance in this area has been the exploitation of compounds containing homometallic bonds as reagents to access heterobimetallic systems. For example, the magnesium dimer $[LMg]_2$ (L = [{(Dipp)NC(CH_3)}_2CH], Dipp = 2,6-iPr_2-C_6H_3) reported by Jones and co-workers[25], and diberyllocene reported by Boronski, Aldridge and co-workers both react with zinc halides to generate new species containing magnesium-zinc[26] and beryllium-zinc[27] bonds, respectively. Heterobimetallic systems can provide significant advances in reactivity, for example [(NON)AlK]_2, reported by Goicoechea, Aldridge and co-workers[28], is a nucleophilic source of

[1]Department of Chemistry, University of Bath, Bath BA2 7AY, UK. [2]Institute for Sustainability, University of Bath, Bath BA2 7AY, UK.
✉e-mail: d.j.liptrot@bath.ac.uk

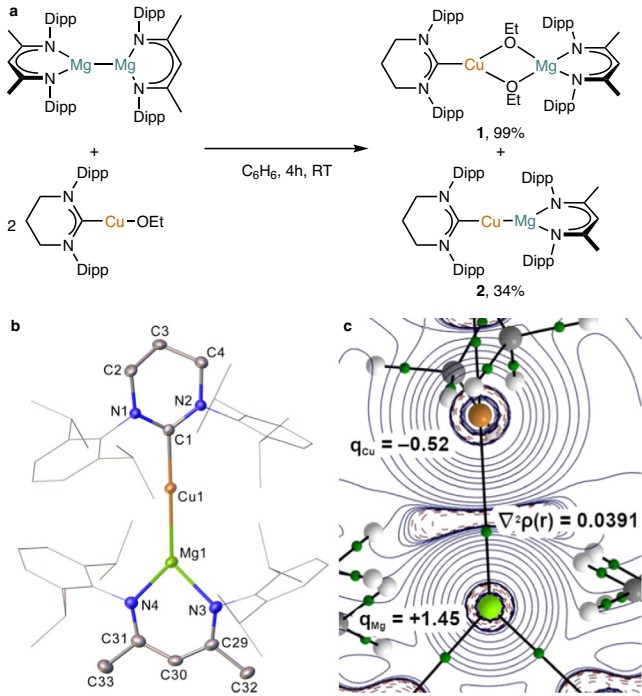

**Fig. 1 | Synthesis, crystallographic, and electronic structure of [(6-Dipp) CuMgL] (2). a** The synthesis of **2** through the reaction of (6-Dipp)CuOEt with [LMg]₂. **b** Solid-state structure of **2**, determined using X-ray crystallography; ADPs displayed at 30% probability; hydrogen atoms omitted for clarity; Diisopropyl-phenyl substituents displayed as wireframes. **c** Laplacian ($\nabla^2\rho(r)$) distribution of the Cu–Mg core of **2**, with the Laplacian at the Cu–Mg bond critical point, and atomic charges labeled, computed at the PBE0-D3BJ/def2-TZVPP//TPSS-D3BJ/def2-SVP level. Dipp = 2,6-iPr₂-C₆H₃.

aluminum such that it (and related species) have stimulated a goldrush in aluminum chemistry since this initial report[29–31]. In this work, we present an isolable compound containing a copper-magnesium bond, [(6-Dipp)CuMgL] (6-Dipp = C{N(Dipp)CH₂}₂CH₂). Computational analysis favors the interpretation that this is a copper compound in a low oxidation state, and its reactions with a wide range of electrophiles provide unambiguous evidence for nucleophilicity at copper.

## Results and discussion

We and others have extensively reported on the exploitation of σ-bond metathesis reactions between diboranes and copper(I) alkoxides to access copper(I) boryls[32–34]. Inspired by this approach, we investigated the reaction of an NHC-copper alkoxide with a magnesium(I) dimer. At room temperature, under an inert atmosphere, reaction of two equivalents (6-Dipp)CuOEt with [LMg]₂ in benzene provided NMR spectroscopic evidence for the formation of two species, both of which contained both the 6-Dipp and L ligands, but only one of which contained ethoxide groups. Solvent extraction of the reaction mixture with pentane permitted these species to be separated, crystallized, and characterized as [(6-Dipp)Cu(μ-OEt)₂MgL] (**1**) and [(6-Dipp)CuMgL] (**2**) (Fig. 1). Compound **2** forms as orange crystals, and comprises a monomeric structure (Fig. 1b) containing a near linear copper geometry (C1-Cu1-Mg1, 177.92(7)°). The Cu-Mg distance in **2** (2.5451(9) Å) correlates to the reported single bond covalent radii of the respective atoms (Pyykkö, 2.51 Å; Cordero et al., 2.73 Å)[35,36]. The Cu1-C1 distance (1.991(2) Å) is long for a copper 6-Dipp complex (example long Cu-C distances (Å): (6-Dipp)Cu{μ-CO}₂WCp*(PEt₃), 1.9616(7)[37]; (6-Dipp) CuB(OC(CH₃)₂)₂, 1.9587(12)[38]). Considered together, these data suggest a low oxidation state copper center, bearing a greater ionic radius than the previously reported (6-Dipp)Cu(I) systems.

Compound **2** was investigated by NMR spectroscopy, with a particular focus on precluding the possibility that it was, in fact, its hydride bridged analogue [(6-Dipp)Cu{μ-H}₂MgL] (**3**). The ¹H NMR spectrum of **2** afforded only the appropriate integrals for the proposed formulation, with no evidence of hydride resonances. Addition of an atmosphere of H₂ to an orange solution of **2** in C₆D₆ resulted in fading of the coloration to tan overnight. The resulting ¹H NMR spectrum showed complete disappearance of the signals assigned to **2**, and appearance of a new set of signals for the 6-Dipp and L ligands, as well as a broad singlet integrating to two protons at 2.47 ppm, which was assigned to the bridging hydrides in **3**. Compound **3** could be crystallized from benzene, in a unit cell with very similar unit cell parameters to those for compound **2**. Compound **3** displayed a shortening (relative to **2**) of the Cu···Mg distance to 2.5183(7) Å. The Cu-C bond length was also notably reduced (1.9385(19) Å) relative to that in **2**. Both dimeric copper and magnesium hydrides have metal-metal distances similar to the sum of the covalent radii[39,40], and an analogous effect on the M·M distance is observed when comparing the Mg(I) compound [L'Mg]₂ to its Mg(II) hydride analogue, [L'MgH]₂ (d(Mg-Mg) = [L'Mg]₂, 2.808(1); [L'MgH]₂, 2.7748(12) Å; L' = [{(2,4,6-(H₃C)₃-C₆H₂)NC(CH₃)}₂CH])[41,42]. Nevertheless, these data were interpreted to authenticate that we had indeed isolated compound **2** (for additional experiments to verify this see SI page 4 and Supplementary Figs. 17, 18 and 37–39).

### Quantum chemical investigations

Quantum chemical calculations were performed on **2** (PBE0-D3BJ/def2-TZVPP//TPSS-D3BJ/def2-SVP level) to elucidate the nature of the copper-magnesium interaction. The crystallographically determined and minimized structures were found to be in good agreement (e.g., d(Cu-Mg) = 2.521 Å as compared with 2.5451(9) Å). Natural bond orbital (NBO) calculations were then employed to examine the charge distribution and bonding in **2**[43]. Natural population analysis (NPA) identified a greater than single population of the Cu 4s orbital (1.15 e⁻), while at magnesium the 3s orbital has a markedly lower population (0.86 e⁻). Alongside a Wiberg bond index (WBI) of 0.69, which is low compared to many metal-metal bonds, these data are consistent with a system with some covalency, with formal oxidation states of copper(-I)/magnesium(II). Natural localized molecular orbital analysis within NBO indicates the Cu–Mg bond comprises a 54% contribution from copper (95% 4s, 4% 3d), and a 39% contribution from magnesium (97% 3s, 2% 3p). The charge distribution in **2** was also probed by quantum theory of atoms in molecules (QTAIM) calculations[44], which reveal significant charge localization at copper, (q_Cu = −0.52, q_Mg = +1.45), reflecting an electron rich copper center and further supporting the assignment of a polarized Cu(-I)-Mg(II) bond in **2**. The density (ρ(r) = 0.0371) Laplacian of the electron density ($\nabla^2\rho(r)$ = 0.0391) and total energy density (H(r) = −0.0090) at the bond critical point indicates a stabilizing interaction between the two centers featuring a significant degree of electrostatic character with charge contraction towards the nuclei (Fig. 1c).

To provide further insight into the electronic nature and reactivity of the Cu–Mg bond in **2**, its reaction with H₂ was interrogated by density functional theory (DFT) (PBE0-D3BJ(CPCM = C₆H₆)/def2-TZVPP//TPSS-D3BJ/def2-SVP). The reaction, which reduces a H₂ molecule to two hydride ligands, proceeds in a single step with a high but accessible barrier of +28.9 kcal mol⁻¹ to exergonically afford [(6-Dipp)Cu(μ-H)₂MgL], **3** (−20.2 kcal mol⁻¹) (Fig. 2). The optimized structure of **3** reflects the shortening of the Cu–Mg distance relative to **2** observed crystallographically (d(Cu–Mg)_comp = **2**, 2.521; **3**, 2.514; d(Cu–Mg)_x-ray = **2**, 2.5451(9); **3**, 2.5183(7) Å).

Further QTAIM calculations were performed at sequential structures along the intrinsic reaction coordinate (IRC) between **2** and **3** via TS(**2**-**3**) (Fig. 2c, see Supplementary Fig. 43 and Supplementary

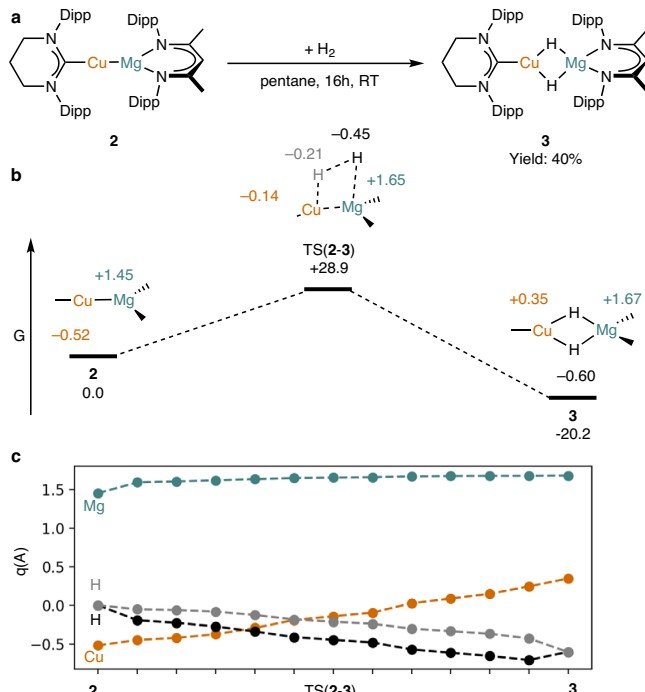

**Fig. 2 | Computational study of the reaction of [(6-Dipp)CuMgL] (2) with H$_2$ to generate [(6-Dipp)Cu(μ-H)$_2$MgL] (3). a** Reaction scheme for the synthesis of **3** from **2** and H$_2$. **b** Computed free energy profile for the reaction (kcal mol$^{-1}$), with structures displaying selected computed atomic charges. **c** Evolution of computed atomic charges across the reaction profile at the Cu (orange), Mg (green) and H (black/gray) centers (PBE0-D3BJ(CPCM = C$_6$H$_6$)/def2-TZVPP//TPSS-D3BJ/def2-SVP level of theory). Dipp = 2,6-iPr$_2$-C$_6$H$_3$.

Table 9). This analysis revealed that as the molecule of H$_2$ initially approaches compound **2** the charge of Mg increases only modestly (**2**: q$_{Mg}$ = +1.45, after initial approach of H$_2$: q$_{Mg}$ = +1.59). Subsequently, a much less pronounced change in charge across the remainder of the reaction surface to **3** is shown (where for **TS**: q$_{Mg}$ = +1.66, **3**: q$_{Mg}$ = +1.68). In contrast, a more consistent and pronounced charge increase is seen at the copper center across the entire reaction surface (where **2**: q$_{Cu}$ = −0.52, **TS**: q$_{Cu}$ = −0.14; **3**: q$_{Cu}$ = +0.35), which is in concert with an increase in the H−H interatomic distance and decreases in charge at each hydrogen center (H$_2$ q$_H$ = 0, **TS**: q$_H$ = −0.21, −0.45, **3**: q$_H$ = −0.60) (Fig. 2). Inspection of the transition state structure is also informative; the copper atom in **2** has a 4s$^2$ 3d$^{10}$ electronic configuration and thus lacks an appropriate orbital to accept electron density from the H$_2$ σ-bond to allow a conventional homolytic cleavage of H$_2$. Instead, the orientation of H$_2$ in TS(**2-3**), and these charge data approximate a synergetic heterolytic hydrogen activation reaction pathway[45]. Initially, the dihydrogen molecule approaches the magnesium center, weakening the Cu−Mg bond and localizing the electron density towards Cu. This is reflected by the slight initial rise in charge at magnesium. The Lewis acidic magnesium center continues to interact with the H$_2$ σ-bond, whilst the electrons in the 4s orbital of the copper atom attack the H$_2$ σ* orbital. These data closely match the oxidative insertion of dihydrogen at nickel(0) complexes with Z-ligands yielding nickel(II) hydride complexes, and are thus consistent with a reaction that most closely approximates an oxidative addition at a copper(−I)/copper(I) pair. Moreover, the orbital picture of this reaction is reminiscent of the FLP-type activation of H$_2$ by moieties with high degrees of charge separation[46–48].

In compound **2**, significant electron density localization at copper can be derived computationally via NBO and QTAIM calculations. Coupled with the relative insensitivity of the charge at magnesium to onward reactivity with hydrogen, this favors the assignment of

**Fig. 3 | Reactivity of compound 2 with dicyclohexylcarbodiimide.** Dipp = 2,6-iPr$_2$-C$_6$H$_3$, Cy = cyclohexyl.

formal oxidation states of Cu(-I)/Mg(II), albeit with the degree of covalency reflecting the formalisms of the oxidation state model[49]. Conjointly, these data reflect a copper-magnesium bond conforming to a Cu$^{δ-}$−Mg$^{δ+}$ charge distribution, i.e., compound **2** is expected to act unambiguously as an copper nucleophile.

### Reactivity of [(6-Dipp)CuMgL], 2

To validate the bond polarization in compound **2** proposed from computation, we investigated its reactivity with a range of substrates containing polar σ- and π-bonds. Addition of dicyclohexylcarbodiimide to a solution of compound **2** in C$_6$D$_6$ led to a discharge of its characteristic orange color over the course of hours. Analysis of the resultant solution by $^1$H NMR spectroscopy indicated complete consumption of the starting materials, and generation of a single major new species. Crystallization indicated this to be the product of a polar π-insertion of the C=N bond into the Cu−Mg bond, [(6-Dipp)CuC(NCy)$_2$MgL] (**4**). This product is the net result of nucleophilic attack of the cupride fragment, [(6-Dipp)Cu]$^−$, on the electrophilic carbon of the carbodiimide (Fig. 3).

In an initial attempt to probe the reactivity of **2** towards polar σ-bonds (Fig. 4), we investigated its reaction with a proton source. Addition of one equivalent of *tert*-butanol to **2** resulted in a $^1$H NMR spectrum containing residual **2**, [(6-Dipp)CuOtBu], and [(6-Dipp)CuH][40]. Adding a second equivalent resulted in loss of the resonances associated with **2** and [(6-Dipp)CuH], but intensified those associated with [(6-Dipp)CuOtBu]. A signal at 4.47 ppm was also observed in both spectra and attributed to dihydrogen. These data allowed us to propose a pathway of reactivity of **2** towards tBuOH. This unselective reaction initially proceeds with protonation of the copper atom in **2**, producing [(6-Dipp)CuH] alongside [LMgOtBu]. [(6-Dipp)CuH] is then a competent base towards the tBuOH present, generating hydrogen gas and [(6-Dipp)CuOtBu]. This net proton reduction, initially to a copper(I) hydride followed by comproportionation of this hydride with a proton to yield hydrogen gas, is noteworthy as it proceeds via the cupride fragment, [(6-Dipp)Cu]$^−$, acting as a Brønsted base (see Supplementary Information page 6 and Supplementary Fig. 36).

Further evidence for the formulation of compound **2** as containing a Cu$^{δ-}$−Mg$^{δ+}$ charge distribution in its central metal-metal bond was provided by replacement of a source of proton by a source of hydride. Thus, addition of PhMe$_2$SiH to **2** in C$_6$D$_6$ followed by overnight heating at 60 °C resulted in near complete consumption of **2**, and the formation of two new species in the $^1$H NMR spectrum, alongside some evidence of decomposition. Comparison of these data to those in the literature facilitated assignment of the magnesium containing species to the hydride dimer, [LMgH]$_2$[50], indicating that the copper containing product was likely to be [(6-Dipp)CuSiMe$_2$Ph], compound **5**. A subsequent synthesis of **5** by adaptation of a literature method permitted confirmation of this thesis. Compound **5** was structurally characterized, and bears metrics that are consistent with its reported analogue, (IPr)CuSiMe$_2$Ph (IPr = C{N(Dipp)CH$_2$})[51].

Upon reaction with other p-block electrophiles, **2** once again acted as a source of [(6-Dipp)Cu]$^−$. Reaction of **2** with chlorobenzene gave rise to peaks we previously attributed to [(6-Dipp)CuPh] in the resultant $^1$H NMR spectrum, and an insoluble precipitate, presumably of [LMgCl][52]. Addition of Ph$_2$POMe to **2** provided data that confirm the

**Fig. 4 | Reactivity of compound 2 with p-block electrophiles.** Dipp = 2,6-iPr₂-C₆H₃, L = [{(Dipp)NC(CH₃)}₂CH].

**Fig. 5 | Reactivity of compound 2 with d-block iodides.** Dipp = 2,6-iPr₂-C₆H₃, L = [{(Dipp)NC(CH₃)}₂CH].

formation of the previously reported copper phosphide, [(6-Dipp)CuPPh₂][53], in the ³¹P NMR spectrum and [LMgOMe] in the ¹H NMR spectrum. In both cases the reactions proceeded within hours to high conversion, and were associated with a loss of the characteristic orange color of **2**.

As well as reacting with hydrogen, proton, and hydride sources, organic substrates and p-block reagents, **2** was competent in transferring the [(6-Dipp)Cu]⁻ synthon to d-block halides (Fig. 5). Addition of **2** to an iron-iodide bond in [CpFe(CO)₂I] provided a ¹H NMR spectrum containing resonances associated with the 6-Dipp and Cp ligands in a 1:1 ratio alongside a precipitate. Synthesis of [(6-Dipp)CuFeCp(CO)₂] via a modified literature route[54] allowed us to assign these resonances to this product, compound **6**, which arises from nucleophilic attack of the copper atom in **2** on iron (Fig. 6a). This occurs with concurrent formation of a ligated magnesium iodide in the form of [LMgI] which precipitates from the reaction. The metric parameters of compound **6** are closely aligned to its previously reported analogue with a 5-membered NHC ligand, [(IPr)CuFeCp(CO)₂], synthesized via the reaction of [(IPr)CuCl] with [CpFe(CO)₂K][54].

Similarly, addition of one equivalent of a zinc iodide, [LZnI], to **2** resulted in formation of a precipitate, and a ¹H NMR spectrum containing, amongst other compounds, data consistent with the formation of a new species containing the 6-Dipp ligand, and an equivalent of L after 4 days. Crystallization from the resultant solution showed this product to be [(6-Dipp)CuZnL], **7**, further validating the interpretation that **2** provides a viable source of nucleophilic copper (Fig. 6b). Compound **7** contains a near linear C-Cu-Zn unit (178.36(6)°) and a Cu-Zn distance (2.3650(4) Å) which is near to the sum of the single bond covalent radii (Pyykkö, 2.30 Å; Cordero et al., 2.54 Å). Compound **7** thus represents structural characterization of an unsupported copper-zinc σ-bond, and is likely to prove an important model system for understanding highly active Cu/ZnO catalysts for carbon dioxide hydrogenation to methanol[55].

Despite its ubiquity in reactivity predicated on positive oxidation states, copper has now been electronically subverted to provide unambiguous nucleophilic character. [(6-Dipp)CuMgL], **2**, is a bona fide source of the cupride fragment, [(6-Dipp)Cu]⁻, towards a wide range of substrates spanning the s-, p-, and d-blocks of the periodic table. Based on the synthesis of [(6-Dipp)CuZnL], **7**, we expect a range of hitherto inaccessible copper complexes to become available through this route, and our current studies involve development of this chemistry.

## Methods
### General considerations and starting materials
All reactions involving air- and moisture-sensitive compounds were carried out under an argon atmosphere using standard Schlenk line and glovebox techniques. NMR experiments using air-sensitive compounds were conducted in J. Young's tap NMR tubes prepared and sealed in a glovebox under argon. All NMR data were acquired at 298 K on an Agilent ProPulse/Bruker Avance NEO instrument for ¹H (500 MHz), ²H (77 MHz) and ¹³C (126 MHz) or Avance NEO 400 instrument for ¹H (400 MHz) and ¹³C (100 MHz). ¹H and ¹³C NMR spectra were referenced using residual C₆D₆ solvent resonances. Data was processed using MestReNova software. Elemental analyses were performed by Elemental Microanalysis Ltd., Okehampton, Devon, U.K. ICP-OES was performed by Butterworth Laboratories Ltd., Teddington, U.K. Benzene was dried over sodium and stored over 4 Å molecular sieves. Hexane and pentane were purified using an MBraun Solvent Purification System and stored over 4 Å molecular sieves. C₆D₆ was dried over a potassium mirror prior to vacuum transfer into a sealed ampoule and stored in the glove box under argon. Starting materials were purchased from standard suppliers and used without further purification unless otherwise stated. [LMg]₂[56], and LZnI[57] were synthesized according to literature conditions.

### Synthesis
**[(6-Dipp)Cu(μ₂-OEt)₂MgL] (1).** In a Schlenk flask to [LMg]₂ (660.0 mg, 1.0 eq, 746.7 μmol) and (6-Dipp)CuOEt (790 mg, 2.1 eq, 1.54 mmol) was added benzene (15 mL) and the resultant slurry stirred. After 2 h additional (6-Dipp)CuOEt (76.7 mg, 0.2 eq, 149.3 μmol) was added and stirring was continued. After two further hours of stirring the volatiles were removed. The resulting crude solid was triturated with pentane (2 × 15 mL). The product was washed twice with pentane and volatiles removed *in vacuo*, yielding a red-gray powder. Retrieved 797.4 mg (99% at 93% purity based on the stoichiometry displayed in Fig. 1). Suitable crystals for SCXRD were grown from a C₆D₆/pentane mixture at −30 °C.

¹H NMR (500 MHz, C₆D₆): δ 7.23 - 7.17 (m, 8H, ArH), 7.03 (d, J = 7.7 Hz, 4H, ArH), 4.92 (s, 1H, NC(CH₃)CH), 3.62 (hept, J = 6.8 Hz, 4H, iPr CH), 3.53 (q, J = 6.8 Hz, 4H, Cu(OCH₂CH₃)₂Mg), 2.89 (hept, J = 7.0 Hz, 4H, iPr CH), 2.65 (t, J = 5.8 Hz, NCH₂CH₂), 1.75 (s, 6H, NC(CH₃)CH), 1.40 - 1.35 (m, 2H, NCH₂CH₂, overlaps with 1.33), 1.33 (d, J = 6.7 Hz, 12H, iPr CH₃), 1.24 (dd, J = 7.0 Hz, 24H, iPr CH₃), 1.10 (d, J = 6.9 Hz, 12H, iPr CH₃), 0.79 (t, J = 6.8 Hz, 6H, Cu(OCH₂CH₃)₂Mg).

¹³C{¹H} NMR (125 MHz, C₆D₆): δ 202.7 (CCu), 167.2 (NC(CH₃)CH), 148.2 (ArC), 145.3 (ArC), 143.1 (ArC), 142.5 (ArC), 129.4 (ArC), 125.2 (ArC), 124.1 (ArC), 123.4 (ArC), 94.6 (NC(CH₃)CH), 60.6 (Cu(OCH₂CH₃)₂Mg), 47.4 (NCH₂CH₂), 28.7 (iPr CH), 28.0 (iPr CH), 25.4 (iPr CH₃), 25.2 (iPr CH₃), 25.1 (iPr CH₃), 24.6 (iPr CH₃), 24.4 (NC(CH₃)CH), 23.9 (Cu(OCH₂CH₃)₂Mg), 20.3 (NCH₂CH₂).

Anal. Calc. for C₆₁H₉₁CuMgN₄O₂: C, 73.25; H, 9.17; N, 5.60%. Found: C, 72.94; H, 9.03; N, 5.59%.

**[(6-Dipp)CuMgL] (2).** In a Schlenk flask to [LMg]₂ (660.0 mg, 1.0 eq, 746.7 μmol) and (6-Dipp)CuOEt (790 mg, 2.1 eq, 1.54 mmol) was added benzene (15 mL) and the resultant slurry stirred. After 2 h additional (6-Dipp)CuOEt (76.7 mg, 0.2 eq, 149.3 μmol) was added and stirring was continued. After two further hours of stirring the volatiles were removed. The resulting crude solid was triturated with pentane (2 × 15 mL). The product was extracted twice with pentane and volatiles removed *in vacuo*. The resulting orange solid is then purified via recrystallization from pentane (229.3 mg, 34% based on the

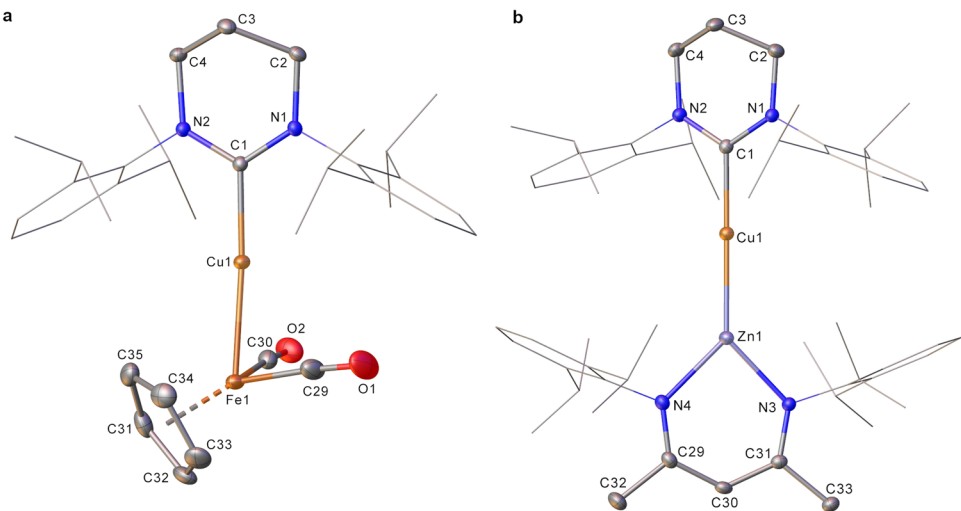

**Fig. 6 | Structural characterization of compounds 6 and 7. a** molecular structure of [(6-Dipp)CuFeCp(CO)$_2$] (compound **6**). **b** Molecular structure of [(6-Dipp)CuZnL] (compound **7**). 30% probability ellipsoids, hydrogen atoms have been omitted. Diisopropylphenyl groups have been represented in wireframe view, for clarity.

stoichiometry displayed in Fig. 1). Suitable crystals for SCXRD were grown from a saturated solution in pentane at −30 °C.

$^1$H NMR (500 MHz, C$_6$D$_6$): δ 7.19 (t, *J* = 7.8 Hz, 2H, Ar$\underline{H}$), 7.13 – 7.10 (m, 4H, Ar$\underline{H}$), 7.0 (d, *J* = 7.8 Hz, 4H, Ar$\underline{H}$), 4.90 (s, 1H, NC(CH$_3$)C$\underline{H}$), 3.13 (hept, *J* = 6.9 Hz, 4H, iPr C$\underline{H}$), 2.89 (hept, *J* = 6.9 Hz, 4H, iPr C$\underline{H}$), 2.55 (t, *J* = 5.9 Hz, 4H, NC$\underline{H_2}$CH$_2$), 1.69 (s, 6H, NC(C$\underline{H_3}$)CH), 1.40 (p, *J* = 6.2 Hz, 2H, NCH$_2$C$\underline{H_2}$), 1.21 (m, 36H, iPr C$\underline{H_3}$), 1.16 (d, *J* = 6.9 Hz, 12H, iPr C$\underline{H_3}$).

$^{13}$C{$^1$H} NMR (125 MHz, C$_6$D$_6$): δ 202.0 ($\underline{C}$Cu), 165.9 (N$\underline{C}$(CH$_3$)CH), 147.0 (Ar$\underline{C}$), 145.3 (Ar$\underline{C}$), 141.8 (Ar$\underline{C}$), 141.5 (Ar$\underline{C}$), 128.8 (Ar$\underline{C}$), 128.6 (Ar$\underline{C}$), 124.4 (Ar$\underline{C}$), 124.3 (Ar$\underline{C}$), 123.5 (Ar$\underline{C}$), 94.6 (NC(CH$_3$)$\underline{C}$H), 47.0 (N$\underline{C}$H$_2$CH$_2$), 28.7 (iPr $\underline{C}$H), 28.3 (iPr $\underline{C}$H), 25.8 (iPr $\underline{C}$H$_3$), 25.8 (iPr $\underline{C}$H$_3$), 24.6 (iPr $\underline{C}$H$_3$), 24.4 (iPr $\underline{C}$H$_3$), 24.0 (NC($\underline{C}$H$_3$)CH), 20.7 (NCH$_2\underline{C}$H$_2$).

IR (cm$^{-1}$) (ATR): 3067, 3031, 2967, 2873, 1523, 1438, 1400, 1319, 1104, 935.2, 846.3, 796.7, 756.2, 377.2.

Anal. Calc. for C$_{57}$H$_{81}$CuMgN$_4$: C, 75.22; H, 8.97; N, 6.16%. Found: C, 74.99; H, 9.42; N, 5.57%.

**[(6-Dipp)Cu{μ-H}$_2$MgL] (3).** In a J. Young's ampoule **2** (100.0 mg, 109.9 µmol) was dissolved in pentane (20 mL). The resulting orange solution was degassed before an atmosphere of H$_2$ was added. Overnight the solution gradually became yellow/tan. The product was filtered, and the solvent was concentrated to ca. 10 mL. The saturated pentane solution was placed stored at −30 °C which over the course of 48 h resulted in crystals that were isolated (40.3 mg, 40%). Suitable crystals for SCXRD were grown from a saturated solution in pentane at −30 °C. Despite repeated attempts, acceptable elemental analysis could not be obtained. This is unsurprising as **3** shows limited stability, in solution it slowly converts to [(6-Dipp)CuH]$_2$ and [LMgH]$_2$. Best attempt: C$_{57}$H$_{83}$CuMgN$_4$: C, 75.05; H, 9.17; N, 6.14%. Found: C, 73.48; H, 8.58, N, 5.27%.

$^1$H NMR (500 MHz, C$_6$D$_6$): δ 7.15 – 7.11 (m, 4H, Ar$\underline{H}$), 7.06 (d, *J* = 7.9 Hz, 4H, Ar$\underline{H}$), 6.93 (d, *J* = 7.7 Hz, 4H, Ar$\underline{H}$), 4.88 (s, 1H, NC(CH$_3$)C$\underline{H}$), 3.18 (hept, *J* = 6.9 Hz, 4H, iPr C$\underline{H}$), 3.07 (hept, *J* = 6.8 Hz, 4H, iPr C$\underline{H}$), 2.74 (t, *J* = 5.9 Hz, 4H, NC$\underline{H_2}$CH$_2$), 2.47 (s, 2H, CuH$_2$Mg), 1.63, (s, 6H, NC(C$\underline{H_3}$)CH), 1.52 (p, *J* = 6.9 Hz 2 H, NCH$_2$C$\underline{H_2}$), 1.24 (d, *J* = 6.8 Hz, 12H, iPr C$\underline{H_3}$), 1.20 (d, *J* = 6.9 Hz, 12H, iPr C$\underline{H_3}$), 1.16 (d, *J* = 6.9 Hz, 12H, iPr C$\underline{H_3}$), 1.13 (d, *J* = 6.9 Hz, 12H, iPr C$\underline{H_3}$).

$^{13}$C{$^1$H} NMR (125 MHz, C$_6$D$_6$): δ 213.0 ($\underline{C}$Cu), 168.5 (N$\underline{C}$(CH$_3$)CH), 145.6 (Ar$\underline{C}$), 145.5 (Ar$\underline{C}$), 142.5 (Ar$\underline{C}$), 142.1 (Ar$\underline{C}$), 128.8 (Ar$\underline{C}$), 128.6 (Ar$\underline{C}$), 124.7 (Ar$\underline{C}$), 124.0 (Ar$\underline{C}$), 94.1 (NC(CH$_3$)$\underline{C}$H), 47.3 (N$\underline{C}$H$_2$CH$_2$), 28.6 (iPr $\underline{C}$H) 28.3 (iPr $\underline{C}$H), 25.6 (iPr $\underline{C}$H$_3$), 25.5 (iPr $\underline{C}$H$_3$), 25.2 (iPr $\underline{C}$H$_3$), 24.4 (iPr $\underline{C}$H$_3$), 24.1 (NC($\underline{C}$H$_3$)CH), 20.8 (NCH$_2\underline{C}$H$_2$).

IR (cm$^{-1}$) (ATR): 3070, 3032, 2967, 2875, 1438, 1402, 1317, 1178, 1105, 934.3, *901.7* (Cu{μ-H}$_2$Mg), 795.6, 757.8, *627.9* (Cu{μ-H}$_2$Mg), 390.4.

For the corresponding deuteride, **3***  ([(6-Dipp)Cu{μ-D}$_2$MgL]), synthesized via repetition of the above with D$_2$ gas.

IR (cm$^{-1}$) (ATR): 3068, 2967, 2875, 1408, 1319, 1178, 1105, 938.2, 796.1, 757.6, *701.1* (Cu{μ-D}$_2$Mg), *500.6* (Cu{μ-D}$_2$Mg), 393.3.

**[(6-Dipp)CuC(NCy)$_2$MgL] (4).** Compound **2** (80.0 mg, 1.0 eq 87.9 µmol) was dissolved in hexane (5 mL) and to it *N,N'*-dicyclohexylcarbodiimide (18.1 mg, 15.73 µL, 1.0 eq, 87.9 µmol) was added. This was allowed to stir overnight. The resulting solution was then filtered and placed in the freezer after which clear crystals were isolated (28.5 mg, 29%). Single crystals suitable for SCXRD were grown from a saturated hexane solution at −30 °C.

$^1$H NMR (500 MHz, C$_6$D$_6$): δ 7.27 – 7.17 (m, 6H, Ar$\underline{H}$), 7.09 (d, *J* = 7.7 Hz, 1H, Ar$\underline{H}$), 7.05 – 7.01 (m, 7H, Ar$\underline{H}$), 4.94 (s, 1H, NC(CH$_3$)C$\underline{H}$), 3.58 (hept, *J* = 6.7 Hz, 4H, iPr C$\underline{H}$), 3.01 – 2.88 (m, 4H, iPr C$\underline{H}$), 2.58 (t, *J* = 5.9 Hz, 4H, NC$\underline{H_2}$CH$_2$), 1.76 (s, 6H, NC(C$\underline{H_3}$)CH), 1.44 (d, *J* = 6.9 Hz, 12H, iPr C$\underline{H_3}$), 1.36 (d, *J* = 6.7 Hz, 12H, iPr C$\underline{H_3}$), 1.33 (d, *J* = 6.8 Hz, 12H, iPr C$\underline{H_3}$), 1.18 (d, *J* = 7.0 Hz, 12H, iPr C$\underline{H_3}$). Proton environments for the cyclohexyl regions were not assigned due to broadening and overlap with the iPr CH$_3$ environments.

$^{13}$C{$^1$H} NMR (125 MHz, C$_6$D$_6$): δ 215.1 (N$_2\underline{C}$Cu), 204.0 ($\underline{C}$Cu), 169.3 (N$\underline{C}$(CH$_3$)CH), 146.8 (Ar$\underline{C}$), 145.9 (Ar$\underline{C}$), 145.1 (Ar$\underline{C}$), 125.2 (Ar$\underline{C}$), 124.9 (Ar$\underline{C}$), 124.6 (Ar$\underline{C}$), 124.0 (Ar$\underline{C}$), 123.8 (Ar$\underline{C}$), 94.6 (NC(CH$_3$)$\underline{C}$H), 47.5 (N$\underline{C}$H$_2$CH$_2$), 36.9 ((Cy-$\underline{C}$H)N), 28.9 (iPr $\underline{C}$H), 26.2 (iPr $\underline{C}$H), 25.2 (iPr $\underline{C}$H$_3$), 24.7 (iPr $\underline{C}$H$_3$), 24.5 (iPr $\underline{C}$H$_3$), 24.1 (NC($\underline{C}$H$_3$)CH), 22.7 (Cy-$\underline{C}$) 20.5 (NCH$_2\underline{C}$H$_2$).

MS (ESI) Expected: 1114.7391, found: 1115.7338 [M + H]$^+$ (err [ppm] = −3.45).

**[(6-Dipp)CuSiMe$_2$Ph] (5).** In a J. Young's NMR tube **2** (14.0 mg, 1.0 eq, 15.4 µmol) was dissolved in C$_6$D$_6$ (0.5 mL). To it was added dimethyl(phenyl)silane (2.37 µL, 2.10 mg, 1.0 eq, 15.4 µmol). The solution was heated to 60 °C for 5 days after which **2** was no longer present in the NMR spectrum and the product, [(6-Dipp)CuSiMe$_2$Ph] (**5**) was identified. 1,3,5-Trimethoxybenzene (9.6 mg) was added as a calibrant to determine the NMR yield of **5** (4.65 mg, 50%).

Authentic synthesis adapted from the synthesis [(IMes)CuSiMe$_2$Ph][58]. To a stirring suspension of (6-Dipp)CuOtBu (0.60 g, 1.0 eq. 1.1 mmol) in toluene (10 mL) was added a solution of PinBSiMe$_2$Ph

(0.34 mL, 1.1 eq, 1.2 mmol) in toluene (30 mL). The resulting dark brown solution was stirred for 18 h, shielded from light. The reaction mixture was filtered, and volatiles were removed *in vacuo* to give solid brown residues. The crude material was washed with hexane (20 mL) and dried *in vacuo* to give 0.44 g of compound **5**. The hexane filtrate was stored at −30 °C overnight, resulting in the crystallization and isolation of a further 0.08 g of **5** as a pale brown crystalline solid (total yield: 0.52 g, 78%). Single crystals suitable for SCXRD were grown from benzene.

$^1$H NMR (500 MHz, C$_6$D$_6$): δ 7.34 − 7.28 (m, 2H, ArH), 7.27 − 7.18 (m, 5H, ArH), 7.09 (d, $J$ = 7.8 Hz, 4H, ArH), 2.99 (hept, $J$ = 6.9 Hz, 4H, iPr CH), 2.68 (t, $J$ = 5.9 Hz, 4H, NCH$_2$CH$_2$), 1.47 (p, $J$ = 6.5 Hz, 2H, NCH$_2$CH$_2$), 1.40 (d, $J$ = 6.9 Hz, 12H, iPr CH$_3$), 1.18 (d, $J$ = 6.9 Hz, 12H, iPr CH$_3$), 0.27 (s, 6H, SiCH$_3$).

$^{13}$C{$^1$H} NMR (125 MHz, C$_6$D$_6$): δ 203.8 (CCu)}, 153.5 (ArC), 145.9 (ArC), 141.3 (ArC), 135.2 (ArC), 129.3 (ArC), 126.9 (ArC), 125.7 (ArC), 124.7 (ArC), 45.8 (NCH$_2$CH$_2$), 28.9 (iPr CH), 25.1 (iPr CH$_3$), 24.7 (iPr CH$_3$), 20.4 (NCH$_2$CH$_2$), 4.3 (SiCH$_3$).

$^{29}$Si NMR (99 MHz, C$_6$D$_6$): δ − 14.3 (SiMe$_2$Ph).

Anal. Calc. for C$_{36}$H$_{51}$CuN$_2$Si: C, 71.65; H, 8.52; N, 4.64. Found: C, 71.31; H, 8.48; N, 4.64.

**[(6-Dipp)CuFeCp(CO)$_2$] (6).** In a J. Young's NMR tube **2** (20.0 mg, 1.0 eq, 22.0 μmol) was dissolved in C$_6$D$_6$ (0.5 mL). To it CpFe(CO)$_2$I (6.7 mg, 1.0 eq, 22.0 μmol) was added. This addition rapidly led to the dissolution of remaining **2** and darkening of the solution. Over a 24-h period the solution became a lighter orange/brown and colorless material precipitated out. 1,3,5-Trimethoxybenzene (12.8 mg) was added as a calibrant to determine the NMR yield of **6** (7.3 mg, 51%).

Authentic synthesis adapted from the synthesis of (IPr) CuFe(CO)$_2$Cp[59]. (6-Dipp)CuCl (157.6 mg, 1.0 eq, 312.9 μmol) was dissolved in THF (20 mL). Separately, KFe(CO)$_2$Cp (67.6 mg, 1.0 eq, 312.9 μmol) was dissolved in THF (20 mL) and the light brown solution cooled to −78 °C via an acetone/dry ice bath. The yellow THF solution containing (6-Dipp)CuCl (157.6 mg, 1 eq, 312.9 μmol) was then added dropwise to the KFe(CO)$_2$Cp solution. The resultant brown solution was stirred overnight, whilst warming to room temperature. The brown solution was pumped down *in vacuo* to dryness, with the brown solid obtained subsequently washed 3 times with hexane (3 × 15 mL). The remaining brown solid was dissolved in toluene (15 mL) and filtered. A layer of hexane (10 mL) was then added to the orange/red solution, which was cooled to −30 °C overnight. Yellow crystals were isolated and dried *in vacuo* (25.4 mg, 13%).

$^1$H NMR (400 MHz, C$_6$D$_6$): δ 7.25 − 7.19 (m, 2H, ArH), 7.14 (d, $J$ = 7.4 Hz, 4H, ArH), 4.04 (s, 5H, CpH), 3.11 (hept, $J$ = 7.0 Hz, 4 H, iPr CH), 2.78 (t, $J$ = 6.0 Hz, 4H, NCH$_2$CH$_2$), 1.61 (d, $J$ = 6.9 Hz, 12H, iPr CH$_3$), 1.52 (p, $J$ = 6.0 Hz, 2H, NCH$_2$CH$_2$), 1.20 (d, $J$ = 6.9 Hz, 12H, iPr CH$_3$).

$^{13}$C{$^1$H} NMR (100 MHz, C$_6$D$_6$): δ 220.6 (FeCO), 198.2 (CCu), 145.7 (ArC), 142.2 (ArC), 129.6 (ArC), 125.1 (ArC), 77.5 (Cp C), 46.5 (NCH$_2$CH$_2$), 29.0 (iPr CH), 25.1 (iPr CH$_3$), 24.6 (iPr CH$_3$), 20.3 (NCH$_2$CH$_2$).

Anal. Calc. for C$_{35}$H$_{45}$CuFeN$_2$O$_2$: C, 65.16; H, 7.03; N, 4.34. Found: C, 65.08; H, 7.02; N, 4.49.

**[(6-Dipp)CuZnL] (7).** Compound **2** (100.0 mg, 1.0 eq 109.9 μmol) was dissolved in hexane (5 mL). Over the course of 48 h, LZnI (67.0 mg, 1.0 eq, 109.9 μmol) was added in 0.25 eq increments. This was allowed to stir for a further 48 h. The resulting orange suspension was filtered and concentrated to dryness *in vacuo*. The product was recrystallized from pentane, yielding orange crystals (10.0 mg, 10%) a small impurity was attributed to compound **2** which co-crystalized. Crystals suitable for SCXRD experiment were grown from a C$_6$D$_6$/pentane mixture at −35 °C.

$^1$H NMR (400 MHz, C$_6$D$_6$): δ 7.20 (d, $J$ = 7.8 Hz, 2H, ArH), 7.14 − 7.10 (m, 4H, ArH), 6.98 (d, $J$ = 7.8 Hz, 4H, ArH), 4.94 (s, 1H, NC(CH$_3$)CH), 3.16 (hept, $J$ = 6.9 Hz, 4H, iPr CH), 2.83 (hept, $J$ = 7.0 Hz, 4H, iPr CH), 2.51 (t, $J$ = 5.7 Hz, 4H, NCH$_2$CH$_2$), 1.70 (s, 6H, NC(CH$_3$)CH), 1.35 (p, $J$ = 5.7 Hz, 2H, NCH$_2$CH$_2$), 1.23 (d, $J$ = 7.0 Hz, 12H, iPr CH$_3$), 1.18 (d, $J$ = 7.0 Hz, 12H, iPr CH$_3$), 1.15 (d, $J$ = 6.9 Hz, 12H, iPr CH$_3$), 1.10 (d, $J$ = 6.9 Hz, 12H, iPr CH$_3$).

$^{13}$C{$^1$H} NMR (100 MHz, C$_6$D$_6$): δ 203.0 (CCu), 163.6, (NC(CH$_3$)CH), 149.0 (ArC), 145.1 (ArC), 141.7 (ArC), 141.4 (ArC), 129.1 (ArC), 124.7 (ArC), 123.7 (ArC), 123.4 (ArC), 94.4 (NC(CH$_3$)CH), 47.3 (NCH$_2$CH$_2$), 28.5 (iPr CH), 28.3 (iPr CH), 26.0 (iPr CH$_3$), 25.8 (iPr CH$_3$), 24.5 (NC(CH$_3$)CH), 24.3 (iPr CH$_3$), 23.7 (iPr CH$_3$), 20.7 (NCH$_2$CH$_2$).

Anal. Calc. for C$_{57}$H$_{81}$CuN$_4$Zn: C, 71.97; H, 8.58; N, 5.89. Found: C, 72.53; H, 8.45; N, 5.28.

ICP-OES on sample of **7** which had been exposed to air and subsequently dissolved in nitric acid, average of two repeats: Cu, 4.95; Zn, 4.36% m/m. Zn:Cu molar ratio, 0.86:1.

## Data availability

Crystallographic data for the structures reported in this Article have been deposited at the Cambridge Crystallographic Data Centre, under deposition numbers CCDC 2363357 (**1**), 2363358 (**2**), 2363359 (**3**), 2363360 (**4**), 2265494 (**5**), 2363361 (**6**), 2363362 (**7**), and 2408960 ([(6-Dipp)CuOEt]). Copies of the data can be obtained free of charge via https://www.ccdc.cam.ac.uk/structures/. Source Data are provided with this manuscript. All other experimental, computational, crystallographic, and spectroscopic data are available in the Supplementary Information. All data are available from the corresponding author upon request. Source data are provided with this paper.

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

## Acknowledgements

DJL thanks the Royal Society for a University Research Fellowship. This research made use of the Anatra High Performance Computing (HPC) Services at the University of Bath. (University of Bath, Research Computing Group, https://doi.org/10.15125/b6cd-s854). We also thank Dr Claire McMullin for use of the AIMAll and NBO software for electronic structure calculations. Funding: Royal Society University Research Fellowship URF\R1\191066 (DJL), EPSRC Standard Research Grant EP/X01181X/1 (SEN).

## Author contributions

Conceptualization: DJL. Data curation: MFM, DJL. Methodology: RAJ, SEN, DJL. Investigation: RAJ, NJE, DJB, TMHD, RSCC, SEN. Visualization: MFM, SEN, DJL. Funding acquisition: DJL. Project administration: DJL. Supervision: MFM, DJL. Writing – original draft: DJL. Writing – review & editing: RAJ, MFM, SEN, DJL.

## Competing interests

The authors declare no competing interests.
