## [Transparent Peer Review file · Nature Communications]

Nucleophilicity at copper(-I) in a compound with a Cu–Mg bond

Corresponding Author: Dr David Liptrot

Version 0:

Reviewer comments:

Reviewer #1

(Remarks to the Author)

See file attached

Reviewer #2

(Remarks to the Author)

Reviewer #3

(Remarks to the Author)

This manuscript reports the synthesis and structure of a new compound featuring a Cu-Mg bond. The reactivity of this compound shows the nucleophilic character of the Cu atom, which is compatible with the formal oxidation states Cu(-I) and Mg(+2). The degree of novelty of the results, particularly the unusual oxidation state for Cu, makes this work suitable for publication in Nature Comm. after several points are addressed by the authors:

A- In p. 2, second paragraph, it is stated that the Cu-Mg distance is larger than the sum of the single bond covalent radii. However, comparison with the empirical atomic radii of Cordero et al. (Dalton Trans. 2008, 2832) indicates the opposite, the Cu-Mg distance (2.5451 Å) is sensibly shorter than the sum of the covalent radii (2.73 Å). Moreover, a search for Mg-Cu distances in the CSD database recovers 18 crystallographically independent values, with an average of 2.73(4) and a minimum of 2.656 Å. This means that the reported compound showcases the shortest Cu-Mg distance reported so far in a molecular compound.

The quite different perspective obtained by the use of two different sets of covalent radii has to do with the fact that Pyykkö's radii come from a limited set of computed molecules which are little representative of Cu and Mg compounds found in real life. Thus, the Mg radii were obtained from calculated distances in MgH₂ and MgMe₂, while that for Cu comes from Cu₂, CuH, CuMe, CuMe₂⁻, AgCu and AuCu. In contrast, the Cordero et al. radii set comes from hundreds of thousands of X-ray structural data using for all elements the abundant E-N distances.

B- In p. 2, last paragraph, "the shortening of the Cu-Mg distance between 2 and 3 can be attributed to the reduced oxidation state of the copper atom in 2)".

I do not think this is a good explanation, because the bonding in the two cases is very different: a two-center, two-electron bond in 2 vs. a four-center, four-electron bond in 3.

C- In p. 3, second paragraph, it is said that the reaction "heterolytically cleaves H₂", but since the two H atoms in 3 occupy equivalent positions, the H-H bond cleavage is homolytic. This should not be confused with the "heterolytic hydrogen activation" adequately mentioned in p. 4.

D- In p. 4, lines 9-10, it is said that "the copper atom in 2 has a d10 electronic configuration", but such an electron configuration is compatible with the Cu(I) oxidation state, not with Cu(-1). A few lines below, there appear to be "electrons in the 4s orbital of the copper atom", both assertions being in contradiction.

E- In p. 4, second paragraph, it is said that electron density localization at copper in 2 can be observed "crystallographically in the long Mg-Cu bond length". I don't see what is the connection between the bond length and the electron localization. Is there any leading reference that shows such a correlation? This apparently unjustified relationship between molecular structure and Cu oxidation state appears also in the Abstract.

F- The clearest conclusion of this paper is the ability of the Cu atom in compound 2 to act as a nucleophile, as stated by the authors in several points of the manuscript. The assignment of a -1 oxidation state to Cu is consistent with such a chemical behavior, but that assignment should be taken carefully and requires further validation with X-ray absorption spectroscopy, for instance. I therefore suggest the authors to be cautious on this issue.

G- In p. 4, last paragraph, the proposed generation of dihydrogen from (6-Dipp)Cu-H and LMgtBuO seems a bit odd, since it implies the formation of a bond between two nucleophilic centers, taking into account that the H must have a hydridic character.

H- In p. 5, last paragraph, "I" appears in several formulae, apparently referring to an iodide ligand, but in "IPr" it might refer to a nitrogen heterocyclic carbene (NHC). This nomenclature should be clarified in this paragraph.

Reviewer #4

(Remarks to the Author)

This manuscript reports the synthesis, analytical characterization, and reactivity of a new nucleophilic Cu(-I) reagent. In my opinion this work is very important and certainly deserves notice by a broad readership.

The manuscript is well-structured, the interpretation of the results is comprehensible and conclusive. The findings are mainly based on single-crystal X-ray structural analyses of the title Cu(-I) compound (2), a synthesis by-product (1) as well as five reaction products (3-7), but are corroborated by several additional characterization data as well as computational studies. All results agree well with the identity of the title compound as a nucleophilic Cu(-I) species.

The X-ray crystallographic data are generally of good quality and the structure refinement was performed precisely. For the hydrido species 3, the high data quality allows for a clean localization of the two hydride H sites from the difference Fourier map, which is not self-evident, in particular in close proximity to a heavy atom such as Cu. The hydride H sites converge to chemically reasonable positions, which definitely helps to prove (together with the IR and NMR spectroscopic data) that 3 is a Cu(+I) hydride while 2 is not. A brief comment about the localization of the hydride H atoms in 3 may be added. In addition, I would not refine the isotropic displacement parameters of the hydrides freely, but set them to e.g. $U_{iso}(H) = 1.2U_{eq}(Cu)$ (-1.20000 in SHELXL).

While the identity of the reaction products 4, 5, and 6 is beyond doubt and demonstrates the nucleophilic reactivity of the title compound 2, there seems to be a serious issue with compound 7. The proposed identity as a Cu-Zn species is based on X-ray crystallography and additional characterization is limited to NMR spectroscopy and elemental analyses (CHN). In the crystal structure model, the R values become significantly better when the putative Zn site is assigned to Cu, that is, it seems more likely that compound 7 is a dinuclear Cu compound rather than a CuZn compound. This would also fit with the low isolated yield of 7. The additional analytical data are not sufficient to distinguish between the CuCu and CuZn species. Consequently, it is highly recommended to add more analytical data to solve this issue, in particular as the authors state that "compound 7 thus represents the first structural characterization of an unsupported copper-zinc σ -bond". For instance, quantification of the Zn content in the product would be suitable. However, I want to point up that even if 7 is not the proposed CuZn species, this would not diminish the importance and originality of the paper significantly.

Some minor language issues:

- Abstract, line 4: "Single crystal X-ray crystallography" is bad language, please replace by "single-crystal X-ray structural analysis" or simply "X-ray crystallography"
- Main, first paragraph, line 14: "stark contrast to"
- Main, second Paragraph, line 10: "bonds, respectively"
- Results, third paragraph, line 8: replace "very comparable" by "very similar"
- Results, Paragraph following Fig. 6, line 5: replace "linear C-Cu-Zn geometry" by "linear C-Cu-Zn unit"
- Synthesis, compound 2, line 6: replace "recrystallisation in pentane" by "recrystallisation from pentane"

Version 1:

Reviewer comments:

Reviewer #1

(Remarks to the Author)

This referee would like to acknowledge the authors for their thorough revisions and for addressing all the comments and suggestions provided by all the referees in the initial review.

The authors have carefully adapted both the manuscript and the supplementary material to incorporate the requested information, which has significantly improved the clarity, completeness, and overall quality of the work. For example, the authors have provided clarifications regarding the methodology, such as the syntheses of complexes 5 and 6; expanded discussions of the results, particularly the reaction of complex 2 with tert-butanol included in the ESI; and enhanced supporting data, including new reaction schemes and improved quality of some NMR spectra. This referee believes that these additional details, provided in response to his feedback, ensure that the manuscript is now more robust and transparent.

This referee now considers the manuscript to be suitable for publication in Nature Communications, congratulations to the authors for such a nice piece of work.

Reviewer #2

(Remarks to the Author)

Reviewer #3

(Remarks to the Author)

The authors have reasonably addressed the minor concerns expressed in my previous report. There is only one very minor point pending:

In the last paragraph of p. 5 we can read «The metric parameters of compound 6 are closely aligned to its previously reported analogue, [(IPr)CuFeCp(CO)₂], synthesized via the reaction of a copper chloride electrophile with [CpFe(CO)₂K]» Where does IPr (propyl iodide, I guess) come from in such a reaction with a "copper chloride electrophile"?

This said, I recommend publication of this manuscript.

Reviewer #4

(Remarks to the Author)

With this version of the manuscript, the authors submitted two new single-crystal X-ray datasets which are both of good quality. From my perspective there are no more arguments against the publication of the manuscript in the current form.

I would like to add some comments on the identity of compound 7 which was a major issue previously. The new single-crystal data agree better with the proposed identity as a CuZn species. The assignment of the metal atom attached to the NHC ligand as Cu by means of electron density is unambiguous. The electron density of the NacNac-coordinate metal atom fits now indeed better with Zn than with Cu, but here the difference is very small in contrast to the other metal site. There might be some latitude in the composition of the crystals, that is, mixed-crystal formation containing varying percentages of the CuZn and CuCu species. The Zn:Cu molar ratio as determined by ICP-OES is indeed significantly smaller than 1, so that the isolated, crystalline solid might contain some impurity of the CuCu species. I do not agree with the authors that "in situ NMR monitoring ... rule out any significant formation of [(6-Dipp)CuCuL]", because the latter species is expected to be paramagnetic and therefore probably not visible in the NMR spectrum (as mentioned by the authors). In addition, the situation in solution and in the isolated solid product is not necessarily identical, as there could be metal-metal exchange events during crystallization/isolation. However, since the provided characterization data of the isolated product (SC-XRD and ICP-OES) definitely support that the proposed CuZn species is the major component in the solid, and the NMR and reactivity studies indicate that the CuZn compound is formed, this effect is negligible in my opinion.

Response to referees for

Nucleophilicity at copper(-I) in a compound with a Cu–Mg bond

We thank all of the referees for their time assessing our manuscript, and are very appreciative of their useful comments. We have now actioned all of their comments, and a line-by-line response describing how we have done so is below. Actioning these comments has done much to improve the quality of the manuscript, and allowed us to ensure that the data submitted fully supports our conclusions and we appreciate the reviewers' input into our work.

Reviewer 1:

In this article, Liptrot and co-workers describe a method to synthesise an heterobimetallic complex featuring a Cu-Mg bond, conforming to a formal oxidation state of Cu(-I). Moreover, the reactivity of this species has been exemplified against a variety of electrophiles, demonstrating the unambiguous nucleophilicity at the Cu(-I) centre and its behaviour as a cupride anion source. As the authors acknowledge in their manuscript, Cu(I) and Cu(II) species have been widely explored and exploited in organometallic chemistry, while lower oxidation states have received less attention so far. In fact, a few examples can be found, as noted by the authors, of in-situ generated 'Cu anions' from Cu(I) species and two equivalents of a suitable reductant, resulting in formally Cu(-I) nucleophilic species [see Synth. Commun. 20, 2711-2721 (1990); Tetrahedron Lett. 34, 3063-3066 (1993)]. However, these studies did not provide further insights of the nature of such species. Other research groups have reported bimetallic M-Cu species (M = B, Al, Ga) that exhibited ambiphilic character [see Angew. Chem. Int. Ed. 60, 14390- 14393 (2021); Chem. Sci. 12, 13458-13468 (2021); Dalton Trans. 51, 3913-3924 (2022)], but this study constitutes a unique example of a species containing a Cu-Mg bond with a Cu(-I) centre that exhibits unambiguous nucleophilic character. This is an interesting fundamental and high impact work, and for these reasons this reviewer recommends its acceptance provided the authors can discuss and comment the following points in more detail:

We thank the reviewer for their positive assessment of the work, and for their suggestions to improve the manuscript.

A) N-Heterocyclic Carbenes (NHCs) are commonly employed ligands for the synthesis of heterobimetallic Al-Cu complexes, and they exhibit different behaviour depending on the electronic properties of the carbene (see Angew. Chem. Int. Ed. 60, 14390-14393), modulating the nucleophilicity/electrophilicity of the metal centre. Similar electronic tuning can be achieved with phosphine ligands (see Chem. Sci. 12, 13458-13468). Do the authors think the same electronic modulation could be achieved in Cu-Mg heterobimetallic complexes? Did the authors attempt to synthesise analogues of compound 2 starting from other NHC-Cu-OR or R₃PCu-OR complexes? It is probably beyond the scope of this publication, but this would be interesting to prove that Cu(-I)-Mg(II) systems always behave as nucleophilic synthons in contrast to (NON)Al-Cu-L complexes.

This is a really interesting question – we've had some luck expanding the range of NHCs which we look forward to reporting in the future, but have had less success thus far with phosphine ligands. Given the long wait for a structurally characterised phosphine supported copper(I) boryl (Organometallics 2017, 36, 24, 4687–4690) we expect this to be a more challenging, but very interesting set of compounds.

B) This reviewer believes the abbreviation 6-Dipp (6-Dipp = C{N(Dipp)CH₂}₂CH₂) can be confusing for the reader, as a different compound is labelled with number 6.

We appreciate this comment from the reviewer, but hope that the consistent use of bold text for compound **6** vs. the standard type face for (6-Dipp) minimises the risk of confusion. To further aid

this, we have ensured that the boldface number **6** is always prefaced by “compound”, and that (6-Dipp) always appears in parentheses.

C) In Figure 1 the structure of complex 1 is not depicted. Despite this compound is defined in the main text as [(6-Dipp)Cu(μ 2-OEt)2MgL] (1), this reviewer thinks it would be helpful for the reader to include a chemical structure. In addition, schemes in general do not contain information about solvent, reaction time, temperature or reaction yield, which should be also added to enhance clarity.

We have now revised figure 1 to include a sketch of compound **1**, and added information about solvent, reaction time, and temperature to all schemes.

D) In page 2, in the paragraph below Figure 2, the authors state: ‘The Cu1-C1 distance (1.991(2) Å) is the longest reported, to date, for a copper 6-Dipp complex’. Authors should include references to support this statement.

We thank the referee for this useful comment, and have added examples of previous long Cu-C distances in 6-Dipp copper complexes.

*E) In order to investigate the reactivity of species 2, the authors selected dicyclohexyl carbodiimide as electrophile obtaining compound 4 after insertion of the C=N bond into the Cu-Mg bond (Figure 3). Reactivity of heterobimetallic M-Cu species with carbodiimide is usually more selective, in general resulting in C=N insertion into the M- Cu bond. Reactivity of Al-Cu complexes towards carbodiimides is usually compared to CO₂, which has a richer reactivity profile (see *Angew. Chem. Int. Ed.* 60, 14390-14393 and *Chem. Sci.* 12, 13458-13468). This reviewer wonders if the authors tested the reactivity of complex 2 with CO₂, as this would result in a direct comparison of its reactivity against other main group-Cu systems.*

We agree with the referee’s comments that CO₂ would give interesting results, and we did indeed investigate its reaction with **2**. This showed evidence of a much richer, and sadly simultaneous and unselective reactivity profile including insertion, ligand activation, deoxygenation, and subsequent onward reactivity of product species. Whilst we look forward to fully investigating this complex reaction in the fullness of time, we believe it is out of scope for this initial communication where our major focus is validating the electronic nature of **2**.

F) Reactivity of 2 towards tBuOH is described in detail in the last paragraph of page 4. This reviewer thinks it would be helpful for the reader to provide a new Scheme depicting the mechanism proposed by the authors as well as the detected and/or proposed intermediates for better clarity. Alternatively, as the reaction is stated to be not clean/selective, this could be described with complete details in the supporting information.

We thank the reviewer for their comments, and have clarified the text in the manuscript. We have also added a description of the experiment along with a scheme, stacked spectra, and an explanation of the reaction and its products to the ESI.

G) The synthesis reported for [(6-Dipp)CuSiMe2Ph] (5) and [(6-Dipp)CuFeCp(CO)2] (6) in the manuscript (pages 8 and 9) correspond to the alternative routes via LCu-OtBu and PinBSiMe2Ph an Cu(I)-X and a nucleophilic Fe(0) salt. These experimental procedures differ from the conditions reported in Figure 4 and Figure 5, where HSiMe2Ph and CpFe(CO)2I were used. These protocols are described in the ESI, but they are relevant to the main text and should therefore be included there.

We thank the referee for this comment and have added the protocols to the manuscript from the ESI.

H) The list of abbreviations for multiplets in NMR spectra is not specified at the beginning of the ESI. In multiple spectroscopic data, heptet are noted as hept, meanwhile in others it appears as septet (sept). I think that this should be clarified and corrected for consistency.

We thank the reviewer for their comment. We have now standardised the naming convention of the multiplets across the ESI and included a table of abbreviations.

I) The integrals in Figure S8 should match the ones reported in the spectroscopic data list.

We thank the reviewer for noticing this discrepancy regarding the starting materials, (6-Dipp)CuOEt, spectroscopic data. The ESI has now been revised to formally include (6-Dipp)CuOEt along with its NMR, SCXRD and EA data.

J) This reviewer suggests the addition of detailed schemes in the ESI depicting reaction conditions, which will help navigating through experimental data.

We thank the reviewer for their comments regarding the inclusion of schemes into the synthetic procedures for clarity. We have now included schemes illustrating key reaction conditions and products and relevant by-products into the ESI.

K) Baseline correction/phasing should be applied to some spectroscopic data (i.e.: Figure S14)

We thank the reviewer for their critical eye of the attached spectra and have applied baseline correction/phasing where appropriate.

L) There are minor typographic mistakes that should be corrected:

General Considerations and Starting Materials, page 6, line 5 -> 13C (125 Hz)

Page 8, line 3 -> -30 oC

Page 9, line 9. The sentence appears to be incomplete: '... to give 0.44g.' Do the authors mean 0.44g of compound 5?

Page 9, line 2 -> '... 0.08g of...'

Page 9, 10 -> C6D6

Page 9, line 23 -> 1H NMR (400 MHz, C6D6): δ 7.25 – 7.19

We thank the reviewer for their careful read of the manuscript, have addressed the typos they flagged and have made the experimental clearer and more precise.

Reviewer 2:

We thank the reviewer for their help in assessing our manuscript.

Reviewer 3:

This manuscript reports the synthesis and structure of a new compound featuring a Cu-Mg bond. The reactivity of this compound shows the nucleophilic character of the Cu atom, which is compatible with the formal oxidation states Cu(-I) and Mg(+2). The degree of novelty of the results, particularly the unusual oxidation state for Cu, makes this work suitable for publication in Nature Comm. after several points are addressed by the authors:

A- In p. 2, second paragraph, it is stated that the Cu-Mg distance is larger than the sum of the single bond covalent radii. However, comparison with the empirical atomic radii of Cordero et al. (Dalton Trans. 2008, 2832) indicates the opposite, the Cu-Mg distance (2.5451 Å) is sensibly shorter than the sum of the covalent radii (2.73 Å). Moreover, a search for Mg-Cu distances in the CSD database

recovers 18 crystallographically independent values, with an average of 2.73(4) and a minimum of 2.656 Å. This means that the reported compound showcases the shortest Cu-Mg distance reported so far in a molecular compound.

The quite different perspective obtained by the use of two different sets of covalent radii has to do with the fact that Pyykkö's radii come from a limited set of computed molecules which are little representative of Cu and Mg compounds found in real life. Thus, the Mg radii were obtained from calculated distances in MgH₂ and MgMe₂, while that for Cu comes from Cu₂, CuH, CuMe, CuMe₂, AgCu and AuCu. In contrast, the Cordero *et al.* radii set comes from hundreds of thousands of X-ray structural data using for all elements the abundant E-N distances.

We thank the referee for their diligent literature searching, and suggestion of an alternative set of radii. We have added the values from the data reported Cordero *et al.* to the manuscript. Whilst comparison with the 18 values for Cu-Mg distances suggested by the referee may be informative, it is complicated by the presence of bridging ligands in these species.

B- In p. 2, last paragraph, "the shortening of the Cu-Mg distance between 2 and 3 can be attributed to the reduced oxidation state of the copper atom in 2)". I do not think this is a good explanation, because the bonding in the two cases is very different: a two-center, two-electron bond in 2 vs. a four-center, four-electron bond in 3.

The referee is correct that such a comparison is complicated by the distinct nature of the bonding, and we have tightened up our discussion of this point.

C- In p. 3, second paragraph, it is said that the reaction "heterolytically cleaves H₂", but since the two H atoms in 3 occupy equivalent positions, the H-H bond cleavage is homolytic. This should not be confused with the "heterolytic hydrogen activation" adequately mentioned in p. 4.

We thank the referee for bringing this to our attention and have adjusted the language accordingly.

D- In p. 4, lines 9-10, it is said that "the copper atom in 2 has a d¹⁰ electronic configuration", but such an electron configuration is compatible with the Cu(I) oxidation state, not with Cu(-I). A few lines below, there appear to be "electrons in the 4s orbital of the copper atom", both assertions being in contradiction.

We apologise to the referee for abbreviating excessively here – we meant simply to discuss that the filled 3d subshell is unable to accept electron density, and hence abbreviated the electronic configuration. We appreciate this being flagged and have revised accordingly.

E- In p. 4, second paragraph, it is said that electron density localization at copper in 2 can be observed "crystallographically in the long Mg-Cu bond length". I don't see what is the connection between the bond length and the electron localization. Is there any leading reference that shows such a correlation? This apparently unjustified relationship between molecular structure and Cu oxidation state appears also in the Abstract.

There is a consistent and established trend that ionic radii of compounds with lower oxidation states are larger than those with higher oxidation states (e.g., Acta Cryst. 1976, A32, 751-767 Cu(I), 91; Cu(II), 87; Cu(III) 68 pm) and deviations from the expected shortening of M-L bonds in the formation of cationic complexes versus their neutral counterparts has been the topic of much discussion because it is contrary to the expected trend (e.g., Chem 2016, 1, 6, 902-920). Nevertheless, combined with the Cordero *et al.* data, and the referee's sage comment below regarding oxidation state, we have removed these claims from both the manuscript and abstract.

F- The clearest conclusion of this paper is the ability of the Cu atom in compound 2 to act as a nucleophile, as stated by the authors in several points of the manuscript. The assignment of a -1

oxidation state to Cu is consistent with such a chemical behavior, but that assignment should be taken carefully and requires further validation with X-ray absorption spectroscopy, for instance. I therefore suggest the authors to be cautious on this issue.

We thank the referee for their note of caution here, and have tried to ensure that we only proposed oxidation state measures as a formalism, e.g.; “Coupled with the relative insensitivity of the charge at magnesium to onward reactivity with hydrogen, this favors the assignment of formal oxidation states of Cu(-I)/Mg(II), albeit with the degree of covalency reflecting the formalisms of the oxidation state model.” Having now clarified the language throughout we hope the referee is happy that we have indeed taken care with our discussion of the oxidation state. We look forward to reporting on the X-ray spectroscopy of this class of compounds in the coming years.

G- In p. 4, last paragraph, the proposed generation of dihydrogen from (6-Dipp)Cu-H and LMgtBuO seems a bit odd, since it implies the formation of a bond between two nucleophilic centers, taking into account that the H must have a hydridic character.

We thank the reviewer for their critical analysis of this reaction. We have now added a scheme, experimental description, explanation of the reaction and how the products form in the ESI under the NMR-scale reactions section. To clarify, we do not contend that a reaction between (6-Dipp)Cu-H and LMgOtBu releases hydrogen. We propose the hydrogen forms via the well-precedented reaction of the (6-Dipp)CuH with tBuOH which occurs concomitantly with the reaction between **2** and tBuOH, thus providing unselective reactivity.

H- In p. 5, last paragraph, "I" appears in several formulae, apparently referring to a iodide ligand, but in "IPr" it might refer to a nitrogen heterocyclic carbene (NHC). This nomenclature should be clarified in this paragraph.

We have now clarified the language in this section to specify that in all cases we are referring to an iodide ligand.

Reviewer 4:

This manuscript reports the synthesis, analytical characterization, and reactivity of a new nucleophilic Cu(-I) reagent. In my opinion this work is very important and certainly deserves notice by a broad readership.

The manuscript is well-structured, the interpretation of the results is comprehensible and conclusive. The findings are mainly based on single-crystal X-ray structural analyses of the title Cu(-I) compound (2), a synthesis by-product (1) as well as five reaction products (3-7), but are corroborated by several additional characterization data as well as computational studies. All results agree well with the identity of the title compound as a nucleophilic Cu(-I) species.

We thank the referee for their kind and positive comments.

The X-ray crystallographic data are generally of good quality and the structure refinement was performed precisely. For the hydrido species 3, the high data quality allows for a clean localization of the two hydride H sites from the difference Fourier map, which is not self-evident, in particular in close proximity to a heavy atom such as Cu. The hydride H sites converge to chemically reasonable positions, which definitely helps to prove (together with the IR and NMR spectroscopic data) that 3 is a Cu(+I) hydride while 2 is not. A brief comment about the localization of the hydride H atoms in 3 may be added. In addition, I would not refine the isotropic displacement parameters of the hydrides freely, but set them to e.g. $U_{iso}(H) = 1.2U_{eq}(Cu)$ (-1.20000 in SHELXL).

While the identity of the reaction products 4, 5, and 6 is beyond doubt and demonstrates the nucleophilic reactivity of the title compound 2, there seems to be a serious issue with compound 7.

The proposed identity as a Cu-Zn species is based on X-ray crystallography and additional characterization is limited to NMR spectroscopy and elemental analyses (CHN). In the crystal structure model, the R values become significantly better when the putative Zn site is assigned to Cu, that is, it seems more likely that compound **7** is a dinuclear Cu compound rather than a CuZn compound. This would also fit with the low isolated yield of **7**. The additional analytical data are not sufficient to distinguish between the CuCu and CuZn species. Consequently, it is highly recommended to add more analytical data to solve this issue, in particular as the authors state that "compound **7** thus represents the first structural characterization of an unsupported copper-zinc σ -bond". For instance, quantification of the Zn content in the product would be suitable. However, I want to point up that even if **7** is not the proposed CuZn species, this would not diminish the importance and originality of the paper significantly.

We thank the referee for this full analysis of our work. We completely agree that the compound better refines as a Cu-Cu system, and appreciate the referee flagging this. Our initial rationale for the assignment of the structure was supported by two additional data points:

- The NMR spectra of **7** show only evidence of diamagnetic species being present. This includes when a calibrant is used to ensure complete signal attribution. [(6-Dipp)CuCuL] would contain an odd number of electrons and would thus likely show paramagnetism, inconsistent with the bulk spectroscopic data for **7**.
- Addition of LZnI to [(6-Dipp)CuZnL] provides spectroscopic data consistent with the formation of [(6-Dipp)CuI] and LZnZnL (see ESI page 8), thus providing evidence that **7** is a source of [LZn], which would not be possible were it [(6-Dipp)CuCuL]

Nevertheless, the referee raises a valid question regarding **7** which we are happy to address in light of the limitations of SC-XRD in differentiating atoms with only one electron difference.

- The low yield of **7** can be readily explained by its high reactivity. *In situ* monitoring of the reaction of **2** with LZnI is informative, and we have added it to the ESI (page 8). This monitoring provides a complete rationale for the low yield of **7**; as noted, **7** is reactive towards LZnI to generate [(6-Dipp)CuI] and LZnZnL. Thus, in an equimolar reaction of **2** and LZnI, a significant volume of the **7** produced reacts with one of the starting materials, thus resulting in a poor yield. Nevertheless, *in situ* NMR monitoring shows that we can adequately trace the fate of the starting materials and products in this reaction to both account for the low yield and rule out any significant formation of [(6-Dipp)CuCuL].
- ICP-OES on an oxidised bulk sample of **7** gives a 1:0.86 molar ratio of copper to zinc. This corresponds to a 93% purity on metals basis of **7**. Combined with the CHN data we thus contend that if there is any [(6-Dipp)CuCuL] present, it represents only a small portion of the sample.
- Finally, we recollected an additional crystal of **7** which provided data consistent with our initial formulation under the code **y24dj110**. We have deposited these data in favour of those previously submitted for CCDC **2363362** (replacing structure **e24dj102** with that of **y24dj110**) for internally consistent analysis on a batch of product synthesised in order to pursue valid referee queries. Our findings are as follows:
 - Copper coordinated to carbene and zinc to NacNac: R1 = 0.0262, wR2 = 0.0739.
 - Copper coordinated to NacNac and zinc to carbene: R1 = 0.0306, wR2 = 0.0833.
 - Copper coordinated to both carbene and NacNac: R1 = 0.0269, wR2 = 0.0744
 - Zinc coordinated to both carbene and NacNac: R1 = 0.0289, wR2 = 0.0833.

We are extremely grateful to the referee for flagging this particular issue. We are now, however, happy that we have provided evidence of both the bulk formulation of compound **7** as being principally comprised of a system containing both Cu and Zn atoms in an appropriate ratio, and that we have selected an appropriate crystal to reflect these bulk data.

Some minor language issues:

Abstract, line 4: "Single crystal X-ray crystallography" is bad language, please replace by "single-crystal X-ray structural analysis" or simply "X-ray crystallography"

The abstract has somewhat changed, but we have incorporated the referee's proposed language.

Main, first paragraph, line 14: "stark contrast to"

Main, second Paragraph, line 10: "bonds, respectively"

Results, third paragraph, line 8: replace "very comparable" by "very similar"

Results, Paragraph following Fig. 6, line 5: replace "linear C-Cu-Zn geometry" by "linear C-Cu-Zn unit"

Synthesis, compound 2, line 6: replace "recrystallisation in pentane" by "recrystallisation from pentane"

We have now incorporated all of these changes, and thank the referee for their careful read of the manuscript.

Reviewer #1

This referee would like acknowledge the authors for their thorough revisions and for addressing all the comments and suggestions provided by all the referees in the initial review.

The authors have carefully adapted both the manuscript and the supplementary material to incorporate the requested information, which has significantly improved the clarity, completeness, and overall quality of the work. For example, the authors have provided clarifications regarding the methodology, such as the syntheses of complexes 5 and 6; expanded discussions of the results, particularly the reaction of complex 2 with tert-butanol included in the ESI; and enhanced supporting data, including new reaction schemes and improved quality of some NMR spectra. This referee believes that these additional details, provided in response to his feedback, ensure that the manuscript is now more robust and transparent.

This referee now considers the manuscript to be suitable for publication in Nature Communications, congratulations to the authors for such a nice piece of work.

We thank the referee for their kind and positive comments.

Reviewer #2

We thank the referee for their assistance in reviewing the manuscript.

Reviewer #3

The authors have reasonably addressed the minor concerns expressed in my previous report. There is only one very minor point pending:

In the last paragraph of p. 5 we can read «The metric parameters of compound 6 are closely aligned to its previously reported analogue, [(IPr)CuFeCp(CO)₂], synthesized via the reaction a copper chloride electrophile with [CpFe(CO)₂K]»

Where does IPr (propyl iodide, I guess) come from in such a reaction with a "copper chloride electrophile"?

This said, I recommend publication of this manuscript.

We thank the referee for their kind and positive comments, and have now revised the section noted to clarify that IPr in this case is a carbene ligand as follows:

The metric parameters of compound 6 are closely aligned to its previously reported analogue with a 5-membered NHC ligand, [(IPr)CuFeCp(CO)₂], synthesized via the reaction of [(IPr)CuCl] with [CpFe(CO)₂K].⁵⁴

Reviewer #4

With this version of the manuscript, the authors submitted two new single-crystal X-ray datasets which are both of good quality. From my perspective there are no more arguments against the publication of the manuscript in the current form.

I would like to add some comments on the identity of compound 7 which was a major issue previously. The new single-crystal data agree better with the proposed identity as a CuZn species. The assignment of the metal atom attached to the NHC ligand as Cu by means of electron density is unambiguous. The electron density of the NacNac-coordinate metal atom fits now indeed better with Zn than with Cu, but here the difference is very small in contrast to the other metal site. There might be some latitude in the composition of the crystals, that is, mixed-crystal formation containing varying percentages of the CuZn and CuCu species. The Zn:Cu molar ratio as determined by ICP-OES is indeed significantly smaller than 1, so that the isolated, crystalline solid might contain some impurity of the CuCu species. I do not agree with the authors that "in situ NMR monitoring ... rule out any significant formation of [(6-Dipp)CuCuL]", because the latter species is expected to be paramagnetic and therefore probably not visible in the NMR spectrum (as mentioned by the authors). In addition, the situation in solution and in the isolated solid product is not necessarily identical, as there could be metal-metal exchange events during crystallization/isolation. However, since the provided characterization data of the isolated product (SC-XRD and ICP-OES) definitely support that the proposed CuZn species is the major component in the solid, and the NMR and reactivity studies indicate that the CuZn compound is formed, this effect is negligible in my opinion.

We thank the referee for their positive and nuanced comments. We agree that a minor amount of another copper-rich component is present in the solid, but that where NMR spectroscopy was used to assess the purity of the CuZn species we added a calibrant to allow us to estimate the amount of NMR silent species in the NMR spectrum. We thus agree with the referee that the major component can now be assigned to the reported species.

In this article, Liptrot and co-workers describe a method to synthesise an heterobimetallic complex featuring a Cu-Mg bond, conforming to a formal oxidation state of Cu(-1). Moreover, the reactivity of this species has been exemplified against a variety of electrophiles, demonstrating the unambiguous nucleophilicity at the Cu(-1) centre and its behaviour as a cupride anion source. As the authors acknowledge in their manuscript, Cu(I) and Cu(II) species have been widely explored and exploited in organometallic chemistry, while lower oxidation states have received less attention so far. In fact, a few examples can be found, as noted by the authors, of in-situ generated 'Cu anions' from Cu(I) species and two equivalents of a suitable reductant, resulting in formally Cu(-1) nucleophilic species [see *Synth. Commun.* **20**, 2711-2721 (1990); *Tetrahedron Lett.* **34**, 3063-3066 (1993)]. However, these studies did not provide further insights of the nature of such species. Other research groups have reported bimetallic M-Cu species (M = B, Al, Ga) that exhibited ambiphilic character [see *Angew. Chem. Int. Ed.* **60**, 14390-14393 (2021); *Chem. Sci.* **12**, 13458-13468 (2021); *Dalton Trans.* **51**, 3913-3924 (2022)], but this study constitutes a unique example of a species containing a Cu-Mg bond with a Cu(-1) centre that exhibits unambiguous nucleophilic character. This is an interesting fundamental and high impact work, and for these reasons this reviewer recommends its acceptance provided the authors can discuss and comment the following points in more detail:

- A) N-Heterocyclic Carbenes (NHCs) are commonly employed ligands for the synthesis of heterobimetallic Al-Cu complexes, and they exhibit different behaviour depending on the electronic properties of the carbene (see *Angew. Chem. Int. Ed.* **60**, 14390-14393), modulating the nucleophilicity/electrophilicity of the metal centre. Similar electronic tuning can be achieved with phosphine ligands (see *Chem. Sci.* **12**, 13458-13468). Do the authors think the same electronic modulation could be achieved in Cu-Mg heterobimetallic complexes? Did the authors attempt to synthesise analogues of compound **2** starting from other NHC-Cu-OR or R₃PCu-OR complexes? It is probably beyond the scope of this publication, but this would be interesting to prove that Cu(-I)-Mg(II) systems always behave as nucleophilic synthons in contrast to (NON)Al-Cu-L complexes.
- B) This reviewer believes the abbreviation 6-Dipp (6-Dipp = C{N(Dipp)CH₂}₂CH₂) can be confusing for the reader, as a different compound is labelled with number 6.
- C) In **Figure 1** the structure of complex **1** is not depicted. Despite this compound is defined in the main text as [(6-Dipp)Cu(μ₂-OEt)₂Mg] (**1**), this reviewer thinks it would be helpful for the reader to include a chemical structure. In addition, schemes in general do not contain information about solvent, reaction time, temperature or reaction yield, which should be also added to enhance clarity.
- D) In page 2, in the paragraph below Figure 2, the authors state: 'The Cu1-C1 distance (1.991(2) Å) is the longest reported, to date, for a copper 6-Dipp complex'. Authors should include references to support this statement.
- E) In order to investigate the reactivity of species **2**, the authors selected dicyclohexyl carbodiimide as electrophile obtaining compound **4** after insertion of the C=N bond into the Cu-Mg bond (Figure 3). Reactivity of heterobimetallic M-Cu species with carbodiimide is usually more selective, in general resulting in C=N insertion into the M-Cu bond. Reactivity of Al-Cu complexes towards carbodiimides is usually compared to CO₂, which has a richer reactivity profile (see *Angew. Chem. Int. Ed.* **60**, 14390-14393 and *Chem. Sci.* **12**, 13458-13468). This reviewer wonders if the authors tested the reactivity of complex **2** with CO₂, as this would result in a direct comparison of its reactivity against other main group-Cu systems.

- F) Reactivity of 2 towards tBuOH is described in detail in the last paragraph of page 4. This reviewer thinks it would be helpful for the reader to provide a new Scheme depicting the mechanism proposed by the authors as well as the detected and/or proposed intermediates for better clarity. Alternatively, as the reaction is stated to be not clean/selective, this could be described with complete details in the supporting information.
- G) The synthesis reported for **[(6-Dipp)CuSiMe2Ph] (5)** and **[(6-Dipp)CuFeCp(CO)2] (6)** in the manuscript (pages 8 and 9) correspond to the alternative routes via LCu-OtBu and PinBSiMe2Ph an Cu(I)-X and a nucleophilic Fe(0) salt. These experimental procedures differ from the conditions reported in Figure 4 and Figure 5, where HSiMe2Ph and CpFe(CO)2I were used. These protocols are described in the ESI, but they are relevant to the main text and should therefore be included there.
- H) The list of abbreviations for multiplets in NMR spectra is not specified at the beginning of the ESI. In multiple spectroscopic data, heptet are noted at hept, meanwhile in others it appears as septet (sept). I think that this should be clarified and corrected for consistency.
- I) The integrals in Figure S8 should match the ones reported in the spectroscopic data list.
- J) This reviewer suggests the addition of detailed schemes in the ESI depicting reaction conditions, which will help navigating through experimental data.
- K) Baseline correction/phasing should be applied to some spectroscopic data (i.e.: Figure S14)
- L) There are minor typographic mistakes that should be corrected:
- General Considerations and Starting Materials, page 6, line 5 -> ^{13}C (125 Hz)
 - Page 8, line 3 -> -30 °C
 - Page 9, line 9. The sentence appears to be incomplete: '... to give 0.44g.' Do the authors mean 0.44g of compound 5?
 - Page 9, line 2 -> '... 0.08g of...'
 - Page 9, 10 -> C₆D₆
 - Page 9, line 23 -> ^1H NMR (400 MHz, C₆D₆): δ 7.25 – 7.19